# Long-Term Trends in PAH Concentrations and Sources at Rural Background Site in Central Europe

**Radek Lhotka [1],\*, Petra Pokorná [2] and Naděžda Zíková [1,2],\***

[1]    Institute for Environmental Studies, Faculty of Science, Charles University in Prague, Albertov 6,
       128 43 Prague, Czech Republic
[2]    Department of Aerosol Chemistry and Physics, Institute of Chemical Process Fundamentals of the Czech
       Academy of Sciences, Rozvojová 1/135, 165 02 Prague, Czech Republic; pokornap@icpf.cas.cz
\*     Correspondence: lhotka@icpf.cas.cz (R.L.); zikova@icpf.cas.cz (N.Z.)

**Abstract:** An increased burden due to polycyclic aromatic hydrocarbons (PAH) is a long-term air quality problem in Central and Eastern Europe. Extensive PAH monitoring has been implemented at the National Atmospheric Observatory Košetice (NAOK), a rural background site in the Czech Republic, as a representative for Central Europe. Data from NAOK are used for evaluation of PAH concentration trends and source apportionment. In total, concentrations of 14 PAHs in particulate matter ($PM_{10}$) and in the gas phase between 2006 and 2016 were evaluated. The highest concentrations were measured at the beginning of the study period in 2006. Mean annual concentrations of benzo(a)pyrene, for example, showed a weak, however statistically significant decreasing trend. The positive matrix factorization (PMF) was used to determine the sources of PAHs at NAOK, with three factors resolved. The probable origin areas of PMF factors were identified by the conditional bivariate probability function (CBPF) and the potential source contribution function (PSCF) methods. The NAOK is affected by local sources of PAHs, as well as by regional and long-range transport. The PAH concentrations correlate negatively with industrial production and traffic intensity. High PAH emissions have been linked to local heating, suggesting that the planned replacement of obsolete combustion sources in the households could improve the overall air quality situation, not only with respect to PAHs.

**Keywords:** $PM_{10}$; polycyclic aromatic hydrocarbons; source apportionment; positive matrix factorization; residential heating

## 1. Introduction

Most polycyclic aromatic hydrocarbons (PAHs) are emitted into the air by anthropogenic sources, as a part of by-products formed by imperfect combustion of hydrocarbon fuels. The major anthropogenic atmospheric emission sources of PAHs include biomass burning, coal and gas combustion, and coke and metal production [1–4]. PAHs are associated with a number of negative impacts on health and on ecosystems. High exposure to some PAHs is associated with an increased risk of DNA adduct formation, reduced mean birth weight of newborns, deteriorated development of nervous pathways, or impairment of human immune system functions [5–9]. A possible carcinogenic effect was observed in connection with eight PAHs [10].

The air quality deterioration due to PAHs (both in the gas- and particulate-phases) still remains an issue in Central and Eastern Europe [11–13]. Directive 2008/50/EC sets the mean annual concentration limit for human health protection to 1 ng·m$^{-3}$ of BaP in $PM_{10}$ [14]. In the Czech Republic, BaP concentrations exceed the limit value on a long-term basis; for example, in 2016, the exceeded limit for BaP was observed in over 26.0% of the country area, corresponding to approx. 61.8% of inhabitants [15].

The overall emissions, including PAHs, are closely related to the economic and social-political situation in individual EU countries, thus associated with industrial production or types of heating in households. The local heating is the main PAHs source in Central Europe [13,16–18].

The emissions of air pollutants ($PM_{10}$, $PM_{2.5}$, and $SO_2$) in Central Europe have declined significantly over the past 20 years. Until the year 2000, these emission reductions led to similar trends in air quality situation in Central European countries. However, since 2000, pollution concentrations in this region have stagnated, only showing limited inter-annual fluctuations [16–18]. Similar trends in concentrations of air pollutants are also observed in the Czech Republic [15,19]. The non-decreasing concentrations of air pollutants are in notable contrast to the continued reductions in air emissions, are not fully understood, and merit detailed examination.

The purpose of this paper is to evaluate the time evolution of concentrations of selected PAHs in particulate matter ($PM_{10}$) and in the gas phase, monitored at a rural background station, the National Atmospheric Observatory Košetice (NAOK) in the period from 1 January 2006 to 31 December 2016, and to estimate PAH sources. Information on changes in concentrations of individual PAHs and in the composition of PAH sources determined during the study period enables us to assess the impact of legislative regulations and economic transformation, in terms of utilization of fuels in households, on final concentrations of these compounds.

## 2. Material and Methods

### 2.1. Experimental

The samples were collected at the National Atmospheric Observatory Košetice, NAOK (N49°35′, E15°05′; 534 m a.s.l., Figure 1) within the monitoring program performed by the Czech Hydrometeorological Institute (CHMI). The NAOK is classified as a rural background site of the Czech Republic. The observatory is quite far from any towns or cities (70 km southeast of Prague and about 20 km from two towns with up to 20,000 inhabitants). However, several small villages are located in the vicinity of the NAOK. One of the major motorways of the Czech Republic lies about 6 km northwest-southeast of the observatory. The NAOK is involved in the EMEP, ACTRIS (formerly EUSAAR) and GAW, and other networks [20].

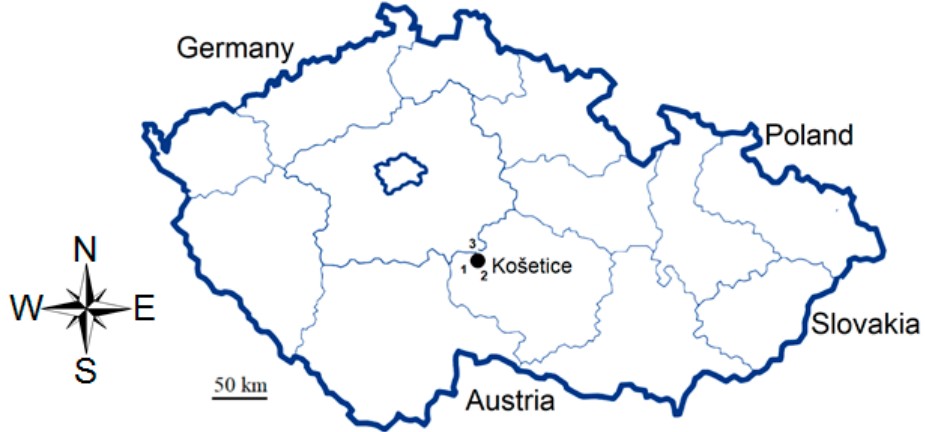

**Figure 1.** Location of the sampling site National Atmospheric Observatory Košetice (NAOK) within the Czech Republic. The location of potential polycyclic aromatic hydrocarbon (PAH) sources cities 1 (Lukavec), 2 (Košetice), and 3 (Čechtice) are indicated in the map [20].

Filters sampling was done every 3 days using LVS3/MVS6 active samplers (Leckel, Germany). The sampling took 24 h with a flow rate of 2.3 $m^3 \cdot h^{-1}$. The samples were collected onto quartz filters (QF) (for sampling particulate PAHs) and polyurethane foam (PUF) discs (for sampling gaseous PAHs). The blanks were used to ensure QA/QC of the entire process [21,22].

The collected samples were analyzed using gas chromatography with mass spectrometry detection (GC/MS) in accordance with the CHMI standard operating procedure [23]. Overall, 14 representatives of PAHs were analyzed: phenanthrene (PHE), anthracene (ANT), fluorene (FLT), fluoranthene (FLA), pyrene (PY), benzo(a)anthracene (BaA), chrysene (CHRY), benzo(b)fluoranthene (BbF), benzo(a)pyrene (BaP), benzo(e)pyrene (BeP), benzo(ghi)perylene (BghiP), indeno(1,2,3-cd) pyrene (IP), dibenzo(a,h)anthracene (DBahA), and benzo(k)fluoranthene (BkF). A more detailed data analysis was focused on the four parameters; FLA—representatives of "light PAHs" and indicators of local combustion effects, BaP—representative of "heavy PAHs" and substance with threshold limit value (TLV), sum of all PAHs (PAH SUM)—indicators of total PAH pollution, and the toxic equivalency factor (TEQ)—a marker of toxic action, calculated according to [24]. The extraction and analytical method of PAHs determination is described in detail in the supplementary information.

Values of routinely monitored meteorological parameters (temperature—T, wind speed—WS, and wind direction—WD) measured 3 times a day (at 7 AM, 2 PM, and 9 PM CET) were assigned to the individual sampling days, and to compare the values to measured PAH concentrations, the mean daily values were used.

## 2.2. PMF Modelling

The positive matrix factorization method EPA PMF 5.0, described in detail in [25], was used to obtain PAH source profiles and their contributions. The data matrix was prepared according to the procedure described in [26]. Values below the reported detection limit (DL) were replaced with DL/2 values, and the value of (5/6) DL * concentration was used as an uncertainty. Measurement uncertainties and detection limits are provided in Table S1. The uncertainty value for PAH SUM (total variable) was determined as 4 * PAH SUM concentration. Seven outlying samples were found in the data matrix and excluded from further analyses (approx. 0.5% samples).

The PMF model was run several times (for different settings of the model) and with different numbers of factors (3–8) in order to obtain a result with the best final diagnostics. Taking into account the distribution of the found residues, variance stability, basic results of the analysis, and evaluation of individual factors, the solution with 3 final factors was chosen.

## 2.3. Wind Analysis

The conditional bivariate probability function method, CBPF, was used to determine the PAH sources. The CBPF method is based on the conditional probability function (CPF) method, with the wind speed added as a third variable and plotted as radial axis values [27]. CBPF is defined as:

$$CBPF_{\Delta\theta,\Delta u} = \frac{m_{\Delta\theta\Delta u} \vdots C \geq x}{n_{\Delta\theta\Delta u}} \quad (1)$$

where $m_{\Delta\theta}$, $\Delta u$ is the number of samples in the wind sector $\Delta\theta$, with the wind speed interval $\Delta u$, where the concentration $C$ is higher than the threshold value $x$; and $n_{\Delta\theta,\Delta u}$ is the total number of samples in this wind speed and direction interval. In this study, the 75th percentile was used as the threshold. The results were processed in R [28] with the openair package [29].

## 2.4. Back Trajectory Analysis

The back trajectories were calculated using the Hybrid Single Particle Lagrangian Integrated Trajectory Model (HYSPLIT) [30], based on the Global Data Assimilation System (GDAS) (1° × 1°) data. The starting height of the trajectories was 500 m (AGL), and the trajectories were calculated for −72 h. The trajectories were generated at 12 PM UTC (i.e., at 1 PM of the local time, CET) and imported to the TrajStat program for calculation of trajectory statistics [31].

The potential source contribution function method, PSCF [32] was used to identify PAH sources. The PSCF expresses the probability that an air mass with the concentration of pollutants higher than a

predetermined criterion reaches the receptor locality after being over a certain geographic area [33]. The PSCF is thus defined as:

$$PSCF_{ij} = \frac{m_{ij}}{n_{ij}} \tag{2}$$

where $n_{ij}$ is the total number of points, and $m_{ij}$ is the number of points with a value higher than the predetermined criterion (75th percentile in our case). In order to prevent inaccurate high values of the *PSCF* due to a low number of end points in some grid cells, the following weight function was applied:

$$W_{i,j} = \begin{cases} 1.0 & 120 < n_{i,j} \\ 0.7 & 60 < n_{ij} \leq 120 \\ 0.4 & 30 < n_{ij} \leq 60 \\ 0.2 & n_{ij} \leq 30 \end{cases} \tag{3}$$

This weighting was used for cells where $n_{ij}$ was lower than 3 times the mean number of end points, i.e., 120 [34].

### 2.5. Trends Calculation

Trends in individual PAH concentrations were evaluated using the Theil-Sen method, based on the non-parametric Mann-Kendall approach [35,36], with mean monthly values or monthly 75th or 95th percentiles taken into account. The average slope is determined by the T parameter:

$$T\left[\%{\cdot}yr^{-1}\right] = 100{\cdot}\left(\frac{C_{End}}{C_{Start}} - 1\right)/N_{years} \tag{4}$$

where $N_{years}$ is the number of years of measurement, and $C_{End}$ and $C_{Start}$ are the mean concentrations for the end and start date, respectively [37,38].

### 2.6. Degree-Days Calculation

The strength of heating seasons was evaluated using the so-called degree-days method. The number of degree-days ($D_{21}$) is determined as follows:

$$D_{21} = (21 - T_{13}) \times N_{13}, \tag{5}$$

where $N_{13}$ is the number of heating days, determined as the number of days in the heating period with the mean temperature lower or equal to 13 °C, and $T_{13}$ is the mean temperature during these days, taken as the mean outdoor temperature [39].

### 2.7. Diagnostic Ratios

The diagnostic ratios (DR) method is used to determine sources of PAHs and is based on the assumption that the monitored PAH sources produce emissions remaining constant in time. In calculating DR, PAHs of the same or similar physics-chemical properties are used. DRs show the ratio of concentrations of individual PAHs or groups of PAHs, for example, A/(A + B) or A/B, where A and B are the concentrations of two different PAHs. Five DRs have been used in this study to determine the sources of PAHs according to [40,41] (Table S2).

## 3. Results and Discussion

### 3.1. Concentrations of Monitored PAHs

The mean annual BaP concentrations and PAH SUM between the years 1996 and 2016 were studied (Table S3). High standard deviation values for all PAHs in all measurement years were caused by high differences in seasonal concentrations of individual PAHs.

The highest PAH SUM concentrations were measured in 1997—36.3 ng·m$^{-3}$. Emissions decreased considerably between 1996 and 1999 (mean concentration of SUM PAH in 1996–1999 was 25.5 ng·m$^{-3}$); however, their decline virtually stopped between 2000 and 2008 (mean concentration of SUM PAH in 2000–2008 was 19.9 ng·m$^{-3}$), and after 2008, the emissions of individual PAHs even started growing (mean concentration of SUM PAH in 2008–2016 was 21.1 ng·m$^{-3}$) [15,19,42]. Mean annual BaP concentrations measured at the NAOK are similar to those found at other background stations in the Czech Republic [15], Germany [43], Austria [44], or Poland [45]. The highest annual mean concentration of BaP was measured at the NAOK in 2006—0.9 ng·m$^{-3}$ (studied period 2006–2016), similarly to other Czech stations, as a result of deteriorated dispersion conditions observed in the given year [15].

Very similar behaviour of mean annual concentrations was found for all 14 PAHs, and the share of individual PAHs on their sum remained virtually unchanged. This indicates that the sources of individual PAHs did not change significantly during the studied period.

## 3.2. Trend Analysis

The trends of individual PAH concentrations for the period from 2006 to 2016 were evaluated using the Theil-Sen method [37,38]. Three statistically significant decreasing trends were found for annual mean concentration: for BaP and TEQ on the significance level $p < 0.05$ (mean annual decrease by 0.01 ng·m$^{-3}$year$^{-1}$), and for PAH SUM on the significance level $p < 0.01$ (mean annual decrease by 0.48 ng·m$^{-3}$year$^{-1}$). The decrease of the PAH concentrations with time was confirmed by a statistically significant Spearman correlation on the $p < 0.05$ significance level (r$_s$, data did not show normal distribution, with alternative hypothesis: true correlation is less than 0) between time and BaP concentrations (r$_s$ = −0.57), TEQ (r$_s$ = −0.60), and PAH SUM (r$_s$ = −0.57).

Statistically significant decreasing trends were also found in the annual 95th percentile concentration (used instead of the annual maximum) for BaP, TEQ, and PAH SUM at the significance level $p < 0.05$. For BaP and TEQ, the decrease was 0.05 ng·m$^{-3}$year$^{-1}$, for PAH SUM 0.15 ng·m$^{-3}$year$^{-1}$.

On the contrary, a decreasing trend, although not statistically significant, was found for the period 1996–2008, according to the study by Klánová et al. [46]. Therefore, the mean annual concentrations of BaP and PAH SUM for the periods from 2006 to 2016 (this study) and from 1996 to 2005 [42] were compared. No statistically significant trend was found for the whole period. However, a statistically significant increasing trend was found for median BaP and FLA concentrations for the cold periods of the year based on data between 1996 and 2016, using the results of Dvorská et al. [47] for the 1996–2005 period (Figure S1). An increase of 0.03 ng·m$^{-3}$year$^{-1}$ (on the significance level $p < 0.05$) was found for BaP medians during the cold periods; for FLA, the increase was 0.16 ng·m$^{-3}$year$^{-1}$ (on the significance level $p < 0.001$). If only the studied period (2006–2016) was taken into account, no statistically significant trend was found for median concentrations during the cold part of the year.

The main source of PAH emissions in Europe [13,16–18] and in the Czech Republic [15,48–51] is local heating, especially stoves heated with solid fuels, such as coal and wood [41]. For example, in 2016, according to CHMI, residential stationary combustion contributed to BaP emissions by 98.3% on the country-wide scale (Figure S2) [15]. Since 2001, the trend of reducing solid fossil fuel consumption in households started weakening or even stopped due to an increase in electricity and gas prices. Although a drop in coal consumption occurred between 2001 and 2008, this decline was compensated by increased firewood consumption. However, this change did not result in pollutant emissions decrease because when wood is combusted, PAHs may be present in high concentrations as well [41]. Solid fuel consumption in households increased in the period of 2009–2012, most likely related to the economic situation (recession). This increase of solid fuel consumption slowed down and finally stopped only after 2013; since then, solid fuel consumption in households has remained unchanged, probably due to stable prices of natural gas and electricity [48–51].

A high share (22%) of households heated by solid fuel boilers was observed in the region of Vysočina, with 40% of households using firewood (at least partially) for heating [52,53]. The portion of households heated by solid fuels near the NAOK is about 30%, i.e., almost a double of

the national average [53]. A persisting high share of old combustion boilers used in households was also observed [52,53]. These old types of boilers have higher emission factors compared to modern boilers [54–56]. An important impact of old combustion boilers on air quality at the NAOK was also confirmed by Schwarz et al. for PM$_{2.5}$ [57].

Additionally, the effect of local heating on the PAH concentrations is evident from the individual PAH concentrations as a function of the heating season intensity, evaluated using the number of degree-days (Figure 2).

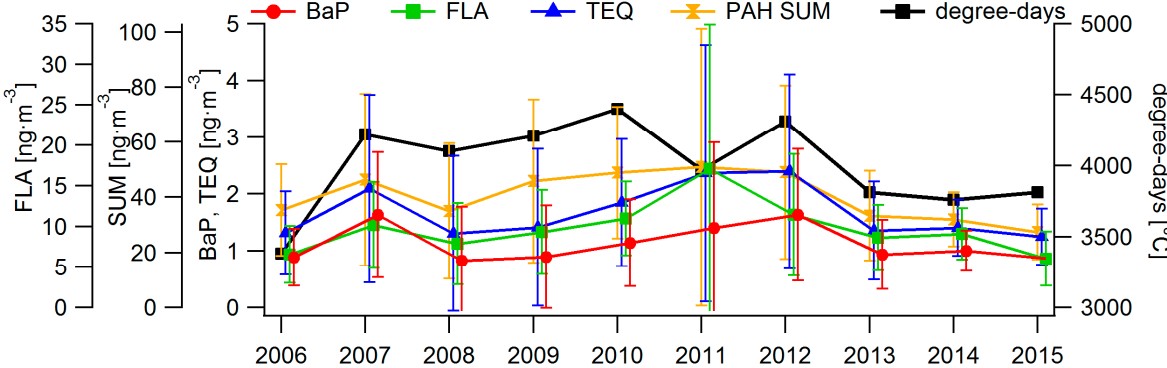

**Figure 2.** The degree-days and medians (± standard deviation) of BaP, TEQ, FLA, and PAH SUM concentrations in the cold half-year (October–March). The year 2006 stands for winter season 2006–2007, etc. The markers are shifted for better readability. FLA—representatives of "light PAHs" and indicators of local combustion effects, BaP—representative of "heavy PAHs" and a substance with threshold limit value (TLV), sum of all PAHs (PAH SUM)—indicators of total PAH pollution, and the toxic equivalency factor (TEQ)—a marker of toxic action, calculated according to [24].

The heating intensity in the studied period was from 3380 to 4400 degree-days, with the season of 2010–2011 being the most intensive heating season (4400 degree-days), followed by the season of 2012–2013 (4310 degree-days), and the season of 2007–2008 (4220 degree-days). During these intensive heating seasons, higher concentration medians were detected at the NAOK for BaP, TEQ, FLA, and PAH SUM than for the rest of the period. The exception is the season 2011–2012 when increased PAH concentrations were measured due to worse dispersion conditions recorded in the Czech Republic despite relatively high temperatures [15,51]. The variation in the dispersion conditions could also be the reason for the high standard deviation measured in this season.

The connection of the PAH concentrations with heating intensity was confirmed by a statistically significant Spearman correlation ($r_s$—alternative hypothesis: true correlation is greater than 0 on the significance level of $p < 0.1$) between PAH SUM concentrations during the heating period and the number of degree-days ($r_s = 0.50$). The number of degree-days correlates also with the 75th percentile of concentration for BaP ($r_s = 0.45$), statistically significant on level of $p < 0.1$, and for PAH SUM ($r_s = 0.66$), for FLA ($r_s = 0.66$), and TEQ ($r_s = 0.62$) on the significance level $p < 0.05$, calculated for the cold period (October to March).

The PAH concentrations correlate negatively with total industrial production, defined by the industrial production index. The industrial production index (IPI), a basic indicator of industrial statistics, is provided by the Czech Statistical Office [58]. The IPI measures industrial economic activities and industry in total, adjusted for price effects [59]. The connection between PAHs and IPI (Figure 3) was confirmed by a statistically significant Spearman correlation ($r_s$, data did not show normal distribution, with the alternative hypothesis: true correlation is greater than 0). For BaP and IPI, a statistically insignificant value $r_s = -0.45$ was found, for PAH SUM and IPI, $r_s = -0.71$ ($p < 0.05$). Overall industrial production was growing steadily between 2000 and 2008, followed by the so-called Great Recession, which lasted until 2013 [59]. The economic production continued to grow again between 2014 and 2017 and continues until today (Figure 3 [60,61]).

Similar relationships as for IPI were also found between PAH concentrations and the traffic intensity (assessed from the fuel consumption, calculated as the sum of all types of fuels, Figure S3 [60]). SUM PAH correlates with traffic with $r_s = -0.90$ ($p < 0.01$), and for BaP it is $r_s = -0.65$ ($p < 0.05$).

The negative correlation coefficient with IPI and traffic intensity indicates that regional sources of PAHs, i.e., the transport and industry in the Czech Republic, are not the only/main sources of PAHs measured at the NAOK and that the studied locality is also affected by long-range transport of PAHs. Similarly, the total fuel consumption is only a proxy for the PAHs emitted from traffic, not taking the car age and/or condition into account, although in Prague, for example, 50% of traffic-related PM emissions originated in the 5% of cars [62]. Both IPI and traffic intensity can be thus used as a proxy for the wealth of the citizens. Assuming that low IPI would result in a higher tendency to use non-typical fuels for household heating (for example domestic waste), the high IPI could result in lower PAH concentrations due to "cleaner" household heating.

PAH concentrations were influenced also by BaP and particulate matter (PM) emissions produced by local heating in the Czech Republic, calculated as the product of consumed fuel amounts and emission factors, characteristic for individual types of boilers [63,64], for 2008–2016 [65]. These parameters show a trend similar to the annual mean PAH concentrations, with Spearman correlation coefficients ($r_s$, data did not show normal distribution) $r_s = 0.52$ between BaP emissions and BaP annual sum ($p < 0.1$, alternative hypothesis: true correlation is greater than 0), $r_s = 0.60$ ($p < 0.05$) for TEQ, $r_s = 0.74$ ($p < 0.01$) for FLA, and $r_s = 0.61$ ($p < 0.05$) for SUM PAH. Correlations were found also between the PM emissions and TEQ ($p < 0.01$, $r_s = 0.81$), FLA ($p < 0.05$, $r_s = 0.70$), and SUM PAH ($p < 0.05$, $r_s = 0.82$). The influence of PM emissions on PAH concentrations was found in the finer resolution as well. PM produced in the area of Bohemian-Moravian Highlands correlates with PAH concentrations with $r_s$ values 0.68, 0.49, and 0.63 for TEQ, FLA, and SUM PAH, respectively. All coefficients are statistically significant at the $p < 0.1$ level (except TEQ with $p < 0.05$).

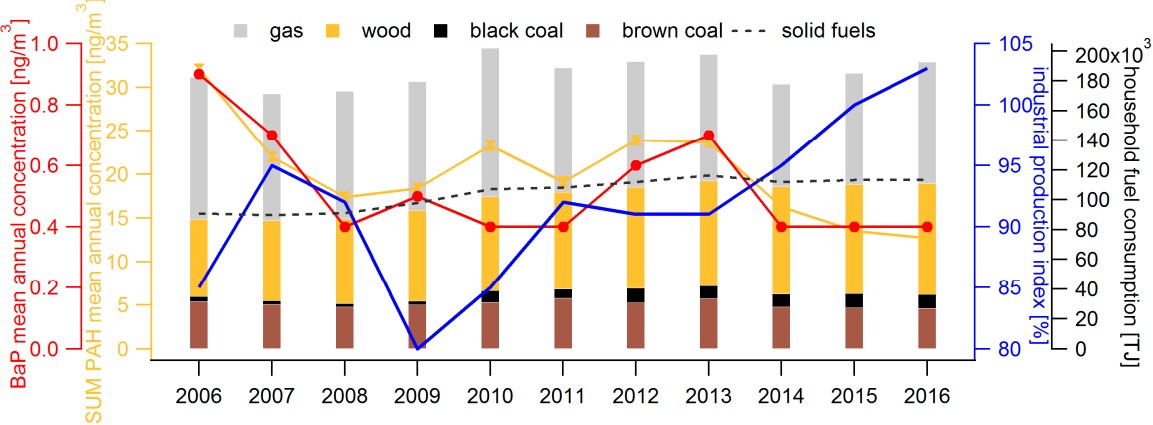

**Figure 3.** Industrial production index, household fuel consumption, and concentration of BaP and SUM PAH/10 in the Czech Republic between 2006 and 2016 (taken from: [61,64]).

No similar relationships were found in comparison with total BaP and PM emissions in the Czech Republic or with BaP and PM emissions in the area of Bohemian-Moravian Highlands resulting from transport or industrial sources [65].

Another influence on PAH concentrations comes from household heating, expressed as the amount of fuels consumed in households between 2006 and 2016 (Figure 3). The mean SUM PAH and TEQ calculated for the cold half of the year correlate (on the significance level of $p < 0.05$) with the ratio of consumption of solid fuels (coal and wood) to the consumption of natural gas, $r_s = -0.60$ and $-0.54$, respectively. The 75th percentile of FLA concentration (calculated for the cold half of the year) correlated with total fuel consumption ($r_s = 0.61$, $p < 0.05$). All evaluated PAH concentrations correlated with the ratio of coal to wood consumption ($r_s$ was 0.60 to 0.94). No statistically significant correlations

were found between PAH concentrations and the amount of consumed fuel for any individual assessed solid fuel (i.e., wood, coal, gas). When evaluating the correlations for the period 2008–2016, more statistically significant relationships between PAH concentrations and fuel consumptions were found. The reason is currently unknown.

High standard deviations were found in concentrations for all four studied compounds (BaP, TEQ, FLA, and PAH SUM) both for mean annual concentration (Table S3) and in the cold half-year concentrations (Figure 2), suggesting high variability in the measured PAH concentrations at the station. Such variability, partly result of variable meteorological conditions, makes any direct conclusion difficult; we have thus tried to evaluate multiple possible parameters (and their time evolution) that could affect the measured PAH concentrations. The relatively small but statistically significant decrease in mean annual BaP, TEQ, and PAH SUM concentrations suggest there is some improvement in the air quality in the Czech Republic. However, the winter concentrations do not show any significant decrease in the studied period (2006–2016), suggesting an unchangeable pattern in the local heating, one of the main sources influencing PAH concentrations at the NAOK.

### 3.3. Annual Cycle of PAH Concentrations

The concentrations of studied PAHs show a characteristic annual cycle, slightly varying over the past 11 years. The highest PAH concentrations were always measured in the cold half-year; however, the position of the maxima differs. The highest concentrations were detected typically in January, in some years also in December (2007 and 2016), February (2012 and 2015), and November (2011) (Figure S4). Different annual cycles of PAH concentrations mainly reflect meteorological (especially mean monthly temperature) and dispersion situation in given year and month. The changes in annual concentrations of the assessed PAHs are thus closely related to better dispersion conditions during cold half-years 2014–2016 compared to 2006–2013. This characteristic mean annual cycle of PAHs (Figure 4) was observed at a number of studied localities (e.g., [66–71]) and is related to seasonality of major PAH sources such as local heating [70], as well as to more frequent occurrences of episodes of deteriorated dispersion conditions in the cold months of the year [72] or reduced photochemical degradation of individual PAHs in the cold part of the year [73].

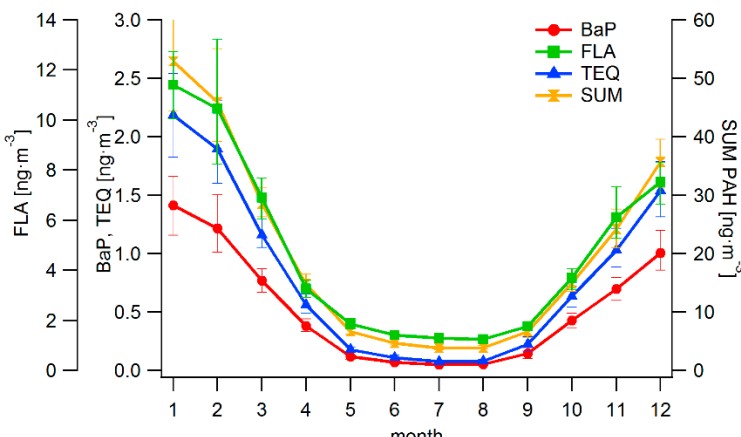

**Figure 4.** Average monthly concentrations (in ng·m$^{-3}$) of 4 selected PAH representatives—BaP, TEQ, FLA, and PAH SUM (curve values) together with 95% confidence intervals for the mean (whiskers).

### 3.4. Identification of PAH Sources Using PMF

The PMF method was used to determine PAH sources at the NAOK. The analysis was done using the data for the warm (April to September) and cold (October to March) period of the year. The three factors result showed to be the best possible solution for both subsets. The details on the diagnostics of the PMF results can be found in Table S4.

Factor one was represented by FLA, PHE, FLT, and PY. This factor can be associated with emissions from coal and firewood combustion; at the NAOK, it most likely originated from industrial facilities in the warm season (Figure 5a) and from local heating in the cold season (Figure 5d) [47,74]. The factor contributes to the PAH SUM by 56% in the warm, and 57% in the cold period of the year.

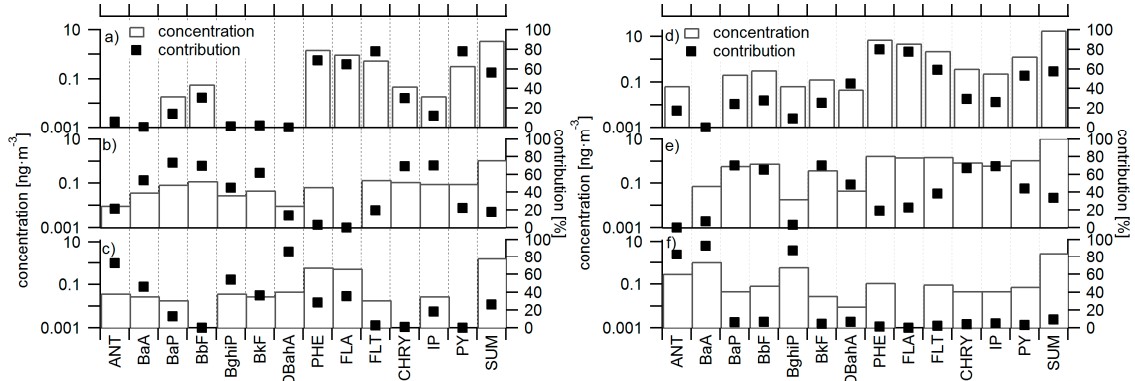

**Figure 5.** The source profiles with bars representing mass contribution (y-axis on the left) and markers representing contribution in percentage (y-axis on the right) in the warm part of the year, (**a**) combustion, (**b**) mixed emissions, (**c**) volatilizing, and in the cold part of the year, (**d**) combustion, (**e**) mixed emissions, (**f**) natural gas combustion.

Factor two was represented by BaP, BbF, BkF, IP, and CHRY. No specific source was found for these compounds. However, a composition similar to this PMF factor was found in urban areas impacted by a number of potential sources including traffic, for example, in Belgium [75] or France [76], therefore, this factor was ascribed to "mixed emissions". The overall contribution of this factor to the PAH SUM was 18% in the warm period (Figure 5b) and 33% in the cold period of the year (Figure 5e). Combustion of organic fuels thus seems to be the main source of these compounds.

Factor 3 was characterized particularly by ANT, BaA, DBahA, and BghiP. This factor represents natural gas and/or light heating oil combustion, either at industrial sources and/or in local heating [77], or alternatively, PAHs volatilizing from the soil [78] or combustion of waste biomass in forests [79]. This is important especially in the warm period of the year when the contribution of this factor to the PAH SUM was 26% (Figure 5c) while reaching only 9% in the cold part of the year (Figure 5f).

For a confirmation, from the PMF factor profiles, DRs were also calculated. Most of the DR values confirm that the combustion factor is affected by solid fuels, mainly coal and biomass combustion, mixed factor by solid fuels combustion and mixed emission (based on DRs with ANT, FEN, BaP, and BghiP), and the volatilizing/combustion factor by combustion in winter and summer by PAH volatilization (based on the same DRs 2) and/or waste biomass combustion (based on DR combining typical combustion PAHS and SUM PAHs, COMB/SUM) (Table S5).

A probable origin of the PMF factors was identified for both periods, warm (Figure 6) and cold (Figure 7).

During the warm parts of the year, potential sources for the combustion factor were situated east and southeast from the study locality. The highest concentrations of this factor were identified for wind speeds over 8 m·s$^{-1}$. Sources located northwest of the study locality were another possible source region for this factor (again with wind speeds over 5 m·s$^{-1}$). The PSCF, however, did not show any probable sources. Potential source areas of the mixed emission factor were similar to the combustion factor. For the volatilizing factor, no specific origin area was found. Thus, in the warm period of the year, the locality was influenced predominantly by distant PAH transport from the east, southeast, and also from the west (also identified, e.g., in [73]). However, no strong potential source areas of individual PMF factors could be identified using the PSCF and CBPF in the warm part of the year (Figure 6).

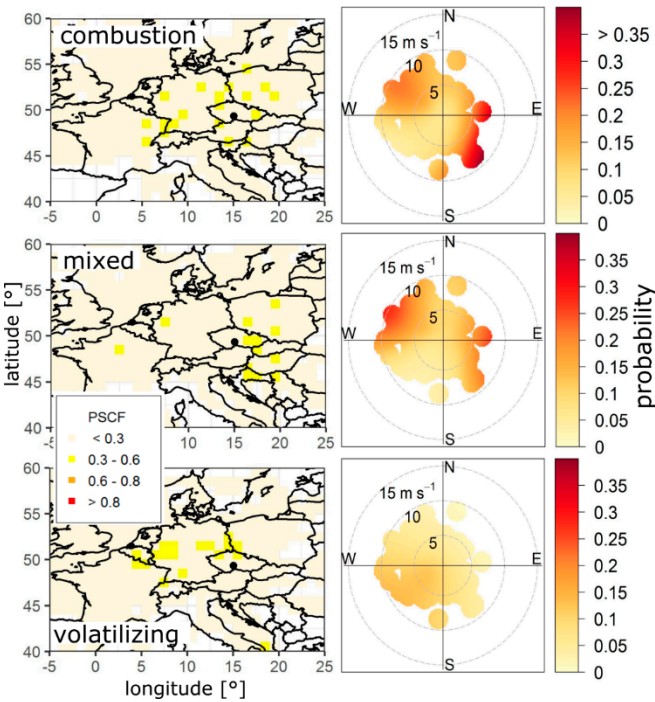

**Figure 6.** Sources of positive matrix factorization (PMF) factors in the warm period of the year, determined using the potential source contribution function (PSCF) method on the left and conditional bivariate probability function (CBPF) method, displayed as probability, on the right.

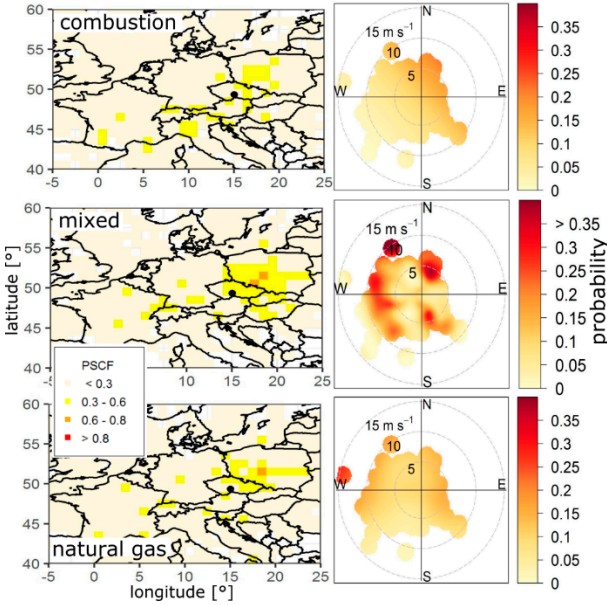

**Figure 7.** Sources of PMF factors in the cold period of the year, determined using the PSCF method on the left and CBPF method, displayed as probability, on the right.

In the cold period of the year (Figure 7), the probability of source location was higher, except for combustion factor. This seems to indicate a good mixing rate of emissions from this source, probably originating in local heating, or alternatively, their dense distribution in the surroundings. For the mixed factor, the potential source locations are partly similar to the natural gas combustion factor, with some locations in southern Poland. The CBPF signal for the mixed factor was found to be stronger from W and N, and a relatively weaker signal from E. High values of the factor were recorded by wind

speeds over 10 m·s$^{-1}$, or, on the other hand, below 5 m·s$^{-1}$, suggesting some additional sources in the vicinity of the NAOK. Natural gas and/or light heating oil combustion factor CBPF plots did not show any strong signals.

The source areas determined using the PSCF method for all three PMF factors in the cold part of the year were similar. An industrial, densely populated area on the border of Moravian-Silesian Region and Poland showed to be an important source region for the individual factors (similar conclusions were also drawn in [63]). High concentrations of pollutants (PM$_{2.5}$ and PAHs) have been detected in a number of studies in this area (e.g., [11,80]).

The results for individual factors also indicate a probable impact of local sources—settlement Košetice in the southeast, Čechtice in the northwest, Lukavec in the west, and other smaller towns in the east (also identified in [56]).

The results of PMF analysis for PAHs determined between 2006 and 2016 are similar to the conclusions found for 1996 to 2009 [46]. The PMF analysis for 1996 to 2009 found three potential PAH sources; however, only two of them were identified (local heating and mixed factor). The determination of possible PM$_{2.5}$ sources (on which PAHs are often bound) was performed at the NAOK [81]. The two of the PMF factors determined in the study indicated combustion sources (local heating and industry). Similar sources were identified southeast, or respectively northeast from the NAOK as well, i.e., the residential heating factor and industrial factors from the east.

The PMF analysis was also done for periods from 2006 to 2010 and from 2011 to 2016 separately to check for changes, as starting from the summer of 2009, some residential fire chambers were modernized, and houses were thermally insulated within the Green Savings program [82]. However, there was no significant change in the modelled PAH sources, and both PMF analyses thus provided almost identical results (differences in the proportions of individual substances in the factors were about 5% to 7%).

The PMF factors were also tested for a trend in concentrations. Based on the Theil-Sen method, no statistically significant trend was found in the mean annual concentrations of individual factors. The same was also applied on the DR values, with a small, however statistically significant trend (+0.01 ng·m$^{-3}$year$^{-1}$ Tables S2 and S5) was found for DR representing solid fuels combustion.

## 4. Conclusions

The study evaluates the trends and sources of PAHs monitored at the NAOK, a rural background site, in the period from January 2006 to December 2016. Overall concentrations of individual PAHs were decreasing very slowly in the study period and depended predominantly on the heating season intensity (fuel composition and consumption amount). Changes occurring in industrial production (e.g., the so-called Great Recession or modernization of industrial plants) and in transport (e.g., adoption of EURO V standard and replacement of vehicles) influenced PAH concentrations; however, the total industrial production and the traffic intensity correlates negatively with the PAH concentrations, as with low industrial production, a higher share of households is expected to use cheaper (and less clean) fuels for heating. Some decrease in the amplitude of annual cycle (winter maxima five to six times higher than summer values between 2006 and 2013, while only double in 2014 and 2016) also collocated with an increased share of renewable energy sources, suggesting a negative effect of combustion on overall PAH concentrations.

The PMF results, combined with diagnostic ratios, confirmed the importance of combustion sources on overall PAH concentrations, particularly in the cold half of the year. At the NAOK, concentrations of individual PAHs are affected by local sources, as well as regional and long-range transport. Source regions for the NAOK include typical hot spot localities of eastern Europe (e.g., Silesian region).

PAHs emissions from local heating have been underestimated in the Czech Republic over a long period. However, replacement of obsolete combustion boilers within the Boiler Subsidies Program (Kotlíkové dotace in Czech) might contribute to an improvement of air quality to a considerable extent.

In addition, operation termination of obsolete combustion systems and their general replacement is planned to be implemented in only a few years ahead. These steps, together with a higher share of renewable energy sources, could improve the overall situation of high ambient PAH concentrations in the Czech Republic.

**Supplementary Materials:** The following are available online at http://www.mdpi.com/2073-4433/10/11/687/s1, Figure S1: The median concentrations (± standard deviation calculated for this study) of BaP, TEQ, FLA and PAH SUM concentrations in the cold half-year (October–March). 2007 stands for winter season 2006–2007 etc. Data till 2006 are from Dvorská at al. [5]. The markers are shifted for a better readability, Figure S2: BaP emission production and concentration of BaP and PAH SUM/10 in the Czech Republic between the years 2006 and 2016 (taken from: [6]), Figure S3: Fuels consumption used for transportation in the Czech Republic between the years 2006 and 2016 (taken from: [7]), Figure S4: Annual cycles of selected PAHs (BaP, TEQ, FLA and PAH SUM) for the years from 2006 to 2016, Table S1: Overall uncertainty and detection limit of measured PAHs (taken from monitoring CHMI), Table S2: Diagnostic ratios, their typical values and characteristic sources (taken from [3,4]), observed trends, together with stat significance level, Table S3: Summary of the mean and standard deviations of BaP and PAH SUM between 1996 and 2016; the BaP and PAH SUM concentrations for 1996–2005 are based on the study of Dvorská et al. [5], Table S4: Summary of PMF diagnostics for PAHs composition, Table S5: Diagnostic ratios for PMF factors for warm (W) and cold (C) part of the year.

**Author Contributions:** Conceptualization N.Z.; Methodology R.L. and N.Z.; analysis R.L. and P.P.; writing R.L., P.P., and N.Z.; supervision, N.Z.

**Funding:** This research was supported by the Ministry of Education, Youth and Sports of the Czech Republic under the grant ACTRIS-CZ LM2015037 and ACTRIS-CZ RI (CZ.02.1.01/0.0/0.0/16_013/0001315).

**Acknowledgments:** We would like to thank the Czech Hydrometeorological Institute for providing meteorological and PAH data, and Milan Váňa and his colleagues (especially Adéla Holubová Šmejkalová) from the NAOK for their valuable cooperation.

**Conflicts of Interest:** The authors declare no conflict of interest.

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
