# Peer review of "Long-Term Trends in PAH Concentrations and Sources at Rural Background Site in Central Europe"

_atmosphere, doi:10.3390/atmos10110687_

Round 1

Reviewer 1 Report

The research was thoroughly carried out with clearly described methodology and comprehensive results discussion. The results are significant from environmental and health point of view. Figures and tables are clear and well present the results. 

However, I would like make the following remarks.

I suggest you don't use the term “immission”. I think that Introduction section could contain slightly more information about PAH, their sources and current state of knowledge in this area. In the abstract, you write that the goal was to analyze PAH in PM10. Meanwhile, in the methodology you describe the intake of PAH on polyurethane foam (PUF) discs and therefore I think that PAH in gas phase was also analyzed. It is therefore unknown whether the described results relate to PAH in PM10 or the sum of particulate and gaseous PAH. This needs to be clarified. You should describe the methodology for extracting PAH from filters and disks in more detail. On page 5 do you mention 14 PAH while in the abstract you indicate 13. Please correct. I think that you should discuss the PAH profile at NAOK more exhaustively. Does it change over the years? Is it similar to the profiles observed for the identified sources of PAH? Maybe it is worth using known diagnostic indicators to verify modeling results? The caption of Figure 4 should contain descriptions of panels a-f.

Reviewer 2 Report

PAH concentrations in PM10 in the Czech Republic measured between 2006 and 2016 are reported, to evaluate the time trend in these concentrations. The PMF technique was used to determine the factors that determine the concentrations, and 3 factors were resolved. The probable sources were investigated using CBPF and PSCF methods. This location is reported to be affected by local, regional, and long-range PAH transport. I suggest the manuscript be accepted after the following specific comments and suggestions are addressed.

Specific comments and suggestions:

The manuscript could benefit from corrections for English language. I have not indicated many of the needed changes in the following. I believe the title should read: “Long-term Trends in PAH Concentrations and Sources at a Rural Background Site in Central Europe” Abstract: For what do CBPF and PSCF stand? Spell out the entire name on first use here. Keywords: Because the PAHs were measured in PM10, I suggest including PM10 instead of PM2.5 in this list. Paragraph below Figure 1: How was concentration computed from these field measurements? Second paragraph below Figure 1: Here is indicated that 14 PAHs were analyzed, whereas the abstract indicates that 13 PAHs were analyzed. Be consistent in these and all other occurrences in the manuscript. Lines following equation 1: Why are the symbols in such tiny font? Second paragraph below Figure 2: This paragraph should offer explanation(s) for why the PAH concentration correlates negatively with total industrial production and traffic intensity. Also, in this paragraph, for what does rs stand? Fourth paragraph in the “Trend Analysis” section, sentence that begins with, “Since 2001,…”: This sentence seems to contradict itself. Can it be written more clearly? Same comment for the last sentence in that paragraph.

Reviewer 3 Report

Mark important PAH source cities on Fig 1 map and add North and South on the map

Put Table 1 in supplement and add standard deviation on Fig 2 points

The whole 3.2 section does not make sense. Although statistical analysis was conducted to explain the trend, Fig 2 clearly shows a fluctuation of the PAH concentrations instead of decreasing trend. Does that mean the slope is too flat to be identified even though it is significant?  A large part of section 3.2 described the Supplemental figures. The figures cannot easily support the result. For example, Fig SI3, the fuel consumption in each category is only listed in the figure providing no sum of the total consumption or correlation with PAH emissions.  Need to rewrite 3.2  section and redo the figures to make it clear and concise.

What does industrial production Index mean?

Minor Changes:

PAHs should be used when describe more than one hydrocarbon

PAHs=polycyclic aromatic hydrocarbons

PAH=polycyclic aromatic hydrocarbon

Abstract; What is “CBPF” and “PSCF”? Add full name or description to explain to readers

Add units in the axis of the graphs

Round 2

Reviewer 3 Report

Section 3.2 needs further improvement.

What is the correlation of coefficient of the PAH concentration with the time? Even though the decrease trend is significant, the correlation can be very weak.

“However, a statistically significant increasing trend was found for mean BaP and FLA concentrations for the cold periods of the year based on data between 1996 and 2016, using the results of Dvorská et al. [47] for the 1996-2005 period.” Please plot the concentration from 1996 to 2016 in the same plot, at least in the supplemental information, for the reader to follow the trend.

The two statements: “ Statistically significant decreasing trends were also found in annual 95th percentile concentration for BaP, TEQ and PAH SUM at the significance level p <0.05. ” and “If only the studied period (2006 - 2016) was taken into account, no statistically significant trend was found for the cold part of the year concentrations.”  are contradictive. Significant or not?

In view of the large standard deviation in Fig 2., I don’t think the discussion on the concentration trends with time in section 3.2 is strong enough. 
